# Current Status and Future Directions of Artificial Intelligence in Post-Traumatic Stress Disorder: A Literature Measurement Analysis

**DOI:** 10.3390/bs15010027

**Published:** 2024-12-30

**Authors:** Ruoyu Wan, Ruohong Wan, Qing Xie, Anshu Hu, Wei Xie, Junjie Chen, Yuhan Liu

**Affiliations:** 1Department of Digital Media Art, School of Architecture and Urban Planning, Huazhong University of Science and Technology, Wuhan 430074, China; u202115710@hust.edu.cn (R.W.); m202474551@hust.edu.cn (W.X.); m202474533@hust.edu.cn (J.C.); 2Academy of Arts & Design, Tsinghua University, Beijing 100084, China; wrh21@mails.tsinghua.edu.cn; 3School of Computer Science and Artificial Intelligence, Wuhan University of Technology, Wuhan 430070, China; felixxq@whut.edu.cn (Q.X.); anshuhu@whut.edu.cn (A.H.); 4MoCT Key Laboratory of Lighting Interactive Service & Tech, Huazhong University of Science and Technology, Wuhan 430074, China

**Keywords:** artificial intelligence, algorithm, bibliometric analysis, Bibliometrix, CiteSpace, digital psychiatry, post-traumatic stress disorder, VOSviewer

## Abstract

This study aims to explore the current state of research and the applicability of artificial intelligence (AI) at various stages of post-traumatic stress disorder (PTSD), including prevention, diagnosis, treatment, patient self-management, and drug development. We conducted a bibliometric analysis using software tools such as Bibliometrix (version 4.1), VOSviewer (version 1.6.19), and CiteSpace (version 6.3.R1) on the relevant literature from the Web of Science Core Collection (WoSCC). The analysis reveals a significant increase in publications since 2017. Kerry J. Ressler has emerged as the most influential author in the field to date. The United States leads in the number of publications, producing seven times more papers than Canada, the second-ranked country, and demonstrating substantial influence. Harvard University and the Veterans Health Administration are also key institutions in this field. The *Journal of Affective Disorders* has the highest number of publications and impact in this area. In recent years, keywords related to functional connectivity, risk factors, and algorithm development have gained prominence. The field holds immense research potential, with AI poised to revolutionize PTSD management through early symptom detection, personalized treatment plans, and continuous patient monitoring. However, there are numerous challenges, and fully realizing AI’s potential will require overcoming hurdles in algorithm design, data integration, and societal ethics. To promote more extensive and in-depth future research, it is crucial to prioritize the development of standardized protocols for AI implementation, foster interdisciplinary collaboration—especially between AI and neuroscience—and address public concerns about AI’s role in healthcare to enhance its acceptance and effectiveness.

## 1. Introduction

### 1.1. Background

#### 1.1.1. Clinical Background of PTSD

According to the World Health Organization (WHO), approximately 3.9% of the global population experiences PTSD at some point in their lives ([43]), with an average of 3.2 traumatic events per person ([48]; [110]). Post-traumatic stress disorder (PTSD) is a severe mental illness characterized by delayed onset, persistence, and heterogeneity. It is prevalent worldwide, especially following exposure to or witnessing extreme distressing events. The latency period of symptoms can range from a few days to several years, and the lifetime prevalence varies by social context and country/region, ranging from 1.3% to 12.2% ([48]). Although 94.4% of people do not develop PTSD after experiencing potentially traumatic events ([48]; [17]), certain types of trauma, such as violent conflict or war, have much higher incidence rates, reaching 15.3% ([32]). Additionally, the prevalence among women is twice that of men (8.6% vs. 4.1%) ([62]; [61]). Many patients rely on external support to manage their symptoms, but PTSD is often misdiagnosed or undiagnosed. Additionally, in low- and middle-income countries, only 25% of PTSD patients report having sought any form of treatment. Stigma and cost are significant barriers to accessing mental health care.

According to the DSM-5 (*Diagnostic and Statistical Manual of Mental Disorders, Fifth Edition*) diagnostic criteria ([17]; [60]; [28]; [92]), PTSD symptoms can be categorized into four types: (i) Traumatic re-experiencing symptoms: Recurring, intrusive memories of the trauma, such as flashbacks or nightmares, cause intense fear and distress, leading to strong emotional or physical reactions. (ii) Avoidance and numbness symptoms: Persistent avoidance of trauma reminders, leading to emotional numbness, detachment, and loss of interest in activities and relationships. (iii) Negative response in cognition and mood symptoms: Ongoing negative emotions like fear, anger, guilt, or shame, often accompanied by anxiety, depression, and, in severe cases, suicidal thoughts or self-harm. (iv) Heightened alertness symptoms: Constant hypervigilance, being easily startled, trouble concentrating, irritability, reckless behavior, and sleep disturbances, including insomnia.

#### 1.1.2. The Impact of Recent Wars and Economic Downturns on PTSD

Localized wars and economic crises worldwide have led to an increase in PTSD prevalence, with multifaceted impacts including increased economic burden, reduced social support, intergenerational effects, and challenges to existing policies and interventions. These factors make PTSD a social issue that requires significant attention and proactive responses. AI intervention presents a crucial opportunity for breakthroughs in early identification and intervention, systematic support, and the provision of personalized treatment plans, especially in the face of growing demands and insufficient societal resources.

The Global Peace Index shows that the level of peace in the world has deteriorated in 12 of the last 16 years, with the level of global peace in 2024 declining by 0.56% from the previous year ([38]). This is well documented, with several hot wars occurring on a small scale around the world as a result of territorial disputes, ethno-religious conflicts, and geopolitical interests, such as the Russia–Ukraine war ([42]), the Syrian Civil War ([2]), the Afghanistan War, the Niger coup, and the Israel–Palestine conflict, all of which have led to a sharp increase in the number of refugees and combatants. The Russia–Ukraine conflict alone has resulted in over 2000 deaths each month over the past two years ([68]), while the Gaza conflict and terrorist attacks in Israel have caused more than 35,000 deaths since October 2023 ([38]), creating severe humanitarian crises. The harshness and duration of these external combat environments, coupled with the frequency of traumatic events and the psychological stress response differences driven by individual genetic factors, can trigger mental disorders among survivors, including homicidal/suicidal ideation, depression, bipolar disorder, PTSD ([42]; [58]; [106]), acute stress reactions, anxiety, adjustment disorders, sleep disorders, and substance abuse ([86]; [4]; [34]). In a 30-year longitudinal study, the point prevalence of PTSD among war survivors was found to be 26.51%, with a high comorbidity with major depression (MD), where 55.26% of those affected by PTSD also suffered from MD ([36]). Survivors and veterans returning to normal society may feel isolated due to a lack of social support and connections, further increasing the risk of PTSD.

Economic downturns, such as those triggered by international financial crises, lead to decreased income, increased debt, rising unemployment rates, increased workload, reduced job stability, and heightened family economic pressure, all of which can negatively impact individual mental health ([109]). These prolonged chronic stressors and anxieties can trigger PTSD-like symptoms, a phenomenon referred to as “Great Depression PTSD”. Additionally, PTSD patients tend to prefer immediate consumption over future savings, making them less capable of coping with economic risks. Surviving refugees may struggle to integrate into new societies and find new paths to survival, while veterans face higher unemployment rates compared to the general population (13.1% vs. 8.5%) due to limited job opportunities ([94]), lower education levels and less work experience, exposing them to greater economic risks.

The impacts of economic crises and localized wars are not limited to the current generation but can have profound effects on subsequent generations. For example, parental PTSD resulting from refugee experiences is associated with mental health disorders in their children, indicating that PTSD can be transmitted across generations. Therefore, it is imperative to accelerate research on AI’s role in the field of PTSD to address issues such as insufficient medical resources, unequal distribution, information asymmetry between patients and healthcare providers, and the difficulties patients face in seeking professional services.

#### 1.1.3. Traditional Diagnostic and Therapeutic Methods and Their Limitations

Traditional PTSD diagnostic approaches mainly include clinical interviews and self-report questionnaires.

Clinical interviews involve face-to-face interactions between a trained therapist and the patient to collect medical history, assess symptoms, comorbidities, and the functional impact on the patient’s psychological state. They are core methods for diagnosing PTSD.

Self-report questionnaires are standardized assessment tools that patients complete based on their feelings and experiences and are used to measure the severity and frequency of PTSD symptoms. These questionnaires rely on patient self-reporting, scoring a series of statements, and are often subject to self-observation bias. Common PTSD self-report questionnaires include the following: (i) Post-Traumatic Stress Disorder Checklist (PCL): a widely used PTSD screening tool with 17 items, available in civilian (PCL-C) and military (PCL-M) versions ([60]; [7]); (ii) Impact of Event Scale-Revised (IES-R): measures trauma-related stress, including intrusive thoughts, avoidance, and hyperarousal; (iii) Beck Depression Inventory (BDI): primarily assesses depressive symptoms, often used to evaluate comorbid depression in PTSD patients; and the (iv) International Classification of Diseases, 11th Revision (ICD-11): issued by the World Health Organization to provide a globally harmonized classification of diseases and health conditions, supporting electronic operation for easy statistics and management ([13]; [66]; [102]).

Traditional PTSD treatments mainly include pharmacotherapy and psychotherapy. Although the Department of Veterans Affairs/Department of Defense (VA/DoD) considers trauma-focused psychotherapy superior to pharmacotherapy ([12]), both are often combined to achieve the best therapeutic outcomes.

Pharmacotherapy includes selective serotonin reuptake inhibitors (SSRIs) such as sertraline, paroxetine, and fluoxetine, which alleviate symptoms by increasing the levels of chemicals in the brain associated with stress and anxiety; benzodiazepines; and the serotonin–norepinephrine reuptake inhibitor (SNRI) venlafaxine ([37]).

Psychotherapy includes:

Cognitive Behavioral Therapy (CBT): This therapy helps patients improve mood and behavior by identifying and challenging distorted thoughts and replacing them with more realistic ones. Techniques such as exposure therapy and cognitive restructuring are integral components of CBT.

Cognitive Processing Therapy (CPT): Developed by Patricia Resick, CPT helps patients challenge negative thoughts and address trauma’s impact on self-image, relationships, and worldview, particularly in areas like safety, trust, control, self-respect, and intimacy.

Prolonged Exposure Therapy (PE): PE reduces fear and avoidance by gradually exposing patients to traumatic memories and triggers. Due to the high involvement of therapists and resource-intensive nature, PE is challenging to implement in remote areas.

Trauma-Focused Cognitive Behavioral Therapy (TF-CBT): EF-CBT employs cognitive and behavioral techniques to modify negative thoughts and behaviors, thereby improving mood and behavior. It is widely used, particularly for children and adolescents.

Eye Movement Desensitization and Reprocessing (EMDR): EMDR helps in reprocessing unprocessed traumatic memories stored in the brain, reducing their negative impact on emotions and daily life.

Additionally, emerging treatments are being researched and applied, such as Emotional Freedom Techniques (EFT) ([24]), Mindfulness-Based Stress Reduction (MBSR), Mindfulness-Based Cognitive Therapy (MBCT), Transcranial Magnetic Stimulation (TMS), and MDMA-assisted therapy.

Despite the availability of effective diagnostic and treatment methods, significant limitations persist. First, subjectivity and bias remain critical issues. Traditional clinical interviews and self-report questionnaires rely heavily on patient self-reporting and clinician interpretation, which can lead to misdiagnosis or underdiagnosis. Second, access to treatment is limited. In resource-poor environments or conflict zones, opportunities to receive evidence-based therapies from trained clinicians are scarce, making timely diagnosis and intervention difficult. Additionally, stigma and cost are barriers that deter people from seeking treatment. There is a need for scientific and visual explanations to help non-professionals (patients and their families) better understand the field. Meanwhile, the development of new drugs has stagnated; since 2006, only a few industry-sponsored PTSD clinical trials have recruited patients ([49]). These limitations highlight the need for innovative approaches, with future research exploring more personalized and advanced diagnostic and treatment methods. AI-driven solutions, for instance, could enhance diagnostic accuracy, increase access to care, and offer more effective personalized treatment options for PTSD.

#### 1.1.4. The Development of AI Technology and Its Role in Mental Health

Artificial intelligence (AI) involves the design of computer algorithms to simulate rational human thinking, enabling machines to perceive, reason, learn, plan, and act to solve complex problems, make decisions, or perform tasks. AI has undergone three developmental stages: (i) the birth of the concept (1956–1966); (ii) the establishment of AI implementation systems (1975–1990), during which machine learning (ML) became central, allowing computers to analyze databases, acquire information, and make intelligent judgments to simulate human experts’ knowledge and experience, assisting doctors in diagnosis and treatment; and (iii) the diversification of AI technology (since the 21st century), which was driven by advancements in algorithms and computer hardware. During this period, scholars have focused on deep learning (DL), which involves algorithms such as convolutional neural networks (CNN) and random forests (RF) to simulate neural networks for deep data analysis and interpretation.

This evolution has positioned AI as an increasingly significant tool in the field of mental health. Through intelligent mental health assessments, early detection, and predictive models, AI provides individuals with more accurate evaluation tools and the ability to identify early signs of mental health issues ([50]; [39]). Additionally, AI can tailor personalized treatment plans based on individual characteristics and offer immediate support through virtual therapists and chatbots, overcoming temporal and geographical barriers. AI’s continuous monitoring capabilities also enable real-time tracking of patients’ mental health status ([53]). However, the application of these technologies raises ethical concerns related to privacy and fairness, which must be carefully addressed as AI continues to develop in this field.

#### 1.1.5. Application and Case Studies of AI in PTSD

The application of AI in the field of PTSD has made significant progress, encompassing various aspects from prediction and diagnosis to treatment and drug development. Below are specific algorithmic logics and case studies, as illustrated in Figure 1.

**(i) Prediction and Prevention:** AI identifies at-risk populations for PTSD and provides early prevention and prediction by analyzing unconventional data such as social media, speech, text, facial expressions, electronic health records, and genetic data ([96]; [72]). Algorithms like Support Vector Machine (SVM), random forest (RF), and logistic regression effectively predict PTSD risk and aid in selecting appropriate interventions ([29]). A specific case involves ML analyzing multivariate predictors to assess PTSD risk among active-duty personnel in Afghanistan ([57]).

**(ii) Diagnosis and Classification:** AI enhances the accuracy and efficiency of PTSD diagnosis by processing imaging data such as magnetic resonance imaging (MRI) ([40]), electroencephalography (EEG), Positron Emission Tomography (PET), and electronic health records (EHRs) ([108]; [71]; [105]). Algorithms like convolutional neural networks (CNN), Riemannian Classifier, Gaussian Process Classifier (GPC), Linear Discriminant Analysis (LDA), and natural language processing (NLP) technologies can identify PTSD types and clarify risk factors. Studies indicate that AI applications in neuroimaging data analysis achieve classification accuracy between 67% and 83.6%. The NYU Langone Health team developed an AI algorithm based on voice analysis with a diagnostic accuracy of 89% for PTSD ([64]; [59]). However, these image analysis approaches and digital decision support systems (DDSS) still require further research due to the challenges posed by small research sample sizes ([9]), a focus on algorithm development, and a lack of real-world interaction ([10]).

**(iii) Treatment and Prognosis Assessment:** AI optimizes treatment plans by integrating multi-source data, improving treatment precision, shortening therapy cycles, and enhancing long-term prognosis ([100]). Utilizing algorithms like Neural Network Classifier (NNC) and Shapley Additive Explanations (SHAP), AI can predict disease progression and treatment outcomes and assist clinicians in developing personalized treatment plans ([77]). AI-assisted psychotherapy software such as Woebot (6.6.0), Tesschatbot, and Wysa (3.7.3) uses NLP to simulate human conversation, analyze patient speech patterns, conduct cognitive therapy, and enable clinicians to remotely monitor patient progress and adjust treatment plans in real time ([33]; [30]). Real-time communication between doctor and patient is facilitated via in-app messaging or video consultations. Additionally, wearable devices such as NightWare, Apollo Neuro, Spire Stone, Biostrap, and Muse Headband ([25]), which combine biofeedback, heart rate monitoring ([80]), and sleep tracking, help monitor and soothe emotions, alleviating PTSD symptoms. Technologies like virtual reality (VR) ([70]), XR, and Augmented Reality (AR) are applied in Virtual Reality Exposure Therapy (VRET) ([99]; [67]), simulating trauma scenarios to help patients safely confront traumatic memories and alleviate symptoms.

**(iv) Self-Management, Real-Time Monitoring, and Remote Intervention:** AI tools monitor patients’ emotions, behavior, and physiological parameters in real time through wearable devices and mobile applications, providing instant emotional support. AI leverages emotion recognition algorithms, reinforcement learning (RL), Mel-frequency cepstral coefficients (MFCCs), and Facial Action Coding System (FACS) technologies to help patients develop personalized self-management plans, preventing disease deterioration ([11]). The PTSD Coach app, jointly created by the National Center for PTSD at the Department of Veterans Affairs and the National Center for Telehealth and Technology at the Department of Defense, features core functions such as symptom tracking, emotion management, crisis support, and educational resources ([85]), aiming to help users manage PTSD symptoms and provide instant support. Crisis Text Line uses AI to analyze users’ text messages, identifying suicide risks in real time and providing emergency interventions. The aforementioned wearable device, Spire Stone, detects negative emotions and suggests relaxation techniques, aiding in the self-regulation of emotions.

**(v) Drug Discovery and Development:** AI identifies molecular pathways and therapeutic targets for PTSD by analyzing data, enabling drug screening, response prediction, and repurposing, thereby improving drug safety and efficacy. AI accelerates new drug development and explores new uses for existing drugs in PTSD treatment by utilizing deep learning, molecular generation models, ensemble learning, NLP, transfer learning, and SVM algorithms. Companies like Insilico Medicine, BenevolentAI, and Atomwise ([5]) leverage DL models, ML models, molecular generation models, and virtual screening platforms ([90]), significantly reducing drug development cycles ([82]).

Despite significant advances in the prediction, diagnosis, treatment, self-management, and drug development for PTSD through AI, many challenges and issues remain to be addressed. Future research must further refine these technologies to enhance their effectiveness and feasibility in clinical practice.

### 1.2. Motivation

#### 1.2.1. Current Limitation of AI’s Role in PTSD

The current state of AI research in the field of PTSD reveals vast potential alongside notable shortcomings.

(i) Data-Related Issues: AI research in PTSD faces several data-related challenges, including data scarcity and incompleteness. Additionally, algorithmic bias, insufficient model generalizability, and the complexity of multimodal data integration are significant hurdles. Given the complexity of AI applications in PTSD, skilled data management personnel are essential to handle large and complex datasets. However, in practice, there is a scarcity of AI scientists who are both capable and willing to collaborate with researchers in the mental health field due to the lack of emphasis on this area and the shortage of talent.

(ii) Clinical Application and Research Needs: While AI has made some clinical advances in imaging data analysis, large-scale controlled trials are still needed to validate its effectiveness and general applicability. For instance, further research is required to determine whether AI-assisted psychiatric treatment is superior to traditional clinical interviews. Current research on acoustic features, EEG ([69]), physiological and biochemical information, and neuroimaging is often fragmented, and integrated applications have not been fully explored, marking a key direction for future studies.

(iii) Legal and Ethical Concerns: The mental health services field already demands careful attention to legal and ethical issues, and the introduction of AI may exacerbate these traditional ethical dilemmas. AI algorithms are typically used in conjunction with big data, raising questions that extend beyond the current legal frameworks. There is ongoing debate about whether AI can be held responsible for ensuring the accuracy and quality of original datasets and the accuracy and validity of the results. Therefore, those involved in the selection, testing, implementation, and evaluation of AI technologies must be acutely aware of these potential ethical issues and prioritize them in decision-making processes.

In conclusion, promoting broad awareness and in-depth research in the field of AI in PTSD is crucial to ensuring the long-term progress of AI technology in mental health.

#### 1.2.2. Significance of Bibliometric Analysis

Bibliometric analysis is a quantitative research method that systematically examines the characteristics and relationships within the scientific literature using statistical and mathematical tools. It allows researchers to objectively evaluate research output, impact, and development trajectories in a specific field, uncovering trends, research hotspots, collaboration networks, and knowledge structures ([74]).

In our study, bibliometric analysis is crucial because it integrates both qualitative and quantitative approaches, enabling the handling of large-scale literature data and producing quantitative results with high-quality visualizations. This method enhances the presentation of AI’s role in PTSD by covering core content, related subfields, and interdisciplinary areas while also offering a detailed discussion of research methods, results, limitations, and future trends. By employing advanced techniques such as co-citation, co-word, and co-authorship network analysis, we provide a comprehensive perspective without sacrificing depth. Furthermore, we ensure transparency and reproducibility by detailing the entire data analysis process, allowing other researchers to replicate and verify our findings.

In the figure below (Figure 2), we compare recent review articles in the same field based on AI algorithms, PTSD pathology, case studies, collaborative network analysis, and the availability of visualized images.

### 1.3. Purpose and Structure of This Study

#### 1.3.1. Purpose of This Study and Related Works and Their Contributions

This study aims to comprehensively review the current status and future prospects of AI in the field of PTSD through bibliometric analysis. Specifically, this study will focus on the following aspects:

(i) Identifying key contributors, institutions, and countries, and their collaborative networks: By analyzing research output, citation frequency, and collaborative networks, this study will identify the leading researchers, academic institutions, and countries involved in applying AI technologies in PTSD and explore the relationships between them, providing insights for future international collaborations and research.

(ii) Extracting valuable insights: Summarizing the specific applications and clinical outcomes of AI technologies (such as machine learning, deep learning, natural language processing, etc.) in the diagnosis, treatment, prediction, and management of PTSD, including their contributions to improving diagnostic accuracy, formulating personalized treatment plans, and enabling remote health monitoring.

(iii) Addressing challenges and proposing solutions based on current technological maturity and clinical development needs: Analyzing key challenges in the application of AI technologies, such as data privacy concerns, algorithmic biases, and difficulties in clinical validation, and proposing corresponding solutions and directions for technological advancement to facilitate the clinical validation and application of AI.

Through these analyses, this study aims to provide theoretical support for the future development of AI interactive tools, promoting early identification of PTSD, personalized treatment plans, and real-time health management, ultimately driving AI technology to better serve clinical practice and the mental health field, offering innovative technological solutions to global mental health challenges.

#### 1.3.2. Article Structure

The structure of this paper can be divided into five sections: The first section provides an overview of PTSD pathology and the foundational role of AI in PTSD, outlining our motivation for writing this paper, including the current research gaps and the necessity of a bibliometric analysis, as well as the objectives of our study. The second section details the research methodology, including data sources and bibliometric analysis methods. The third section presents the results of the analysis, supported by extensive visual mapping. The fourth section offers a discussion, summarizing the findings, analyzing the significance and reasons behind specific results, addressing the limitations of the paper, and providing a future outlook for the field. The fifth section concludes this paper.

## 2. Materials and Methods

### 2.1. Data Source

The data utilized for this analysis were sourced from the Web of Science Core Collection (WoSCC) database, comprising 520 articles retrieved on 25 July 2024, using the following search string: (TS = (“artificial intelligence” OR AI OR “machine learning” OR “deep learning” OR “neural networks” OR “natural language processing” OR “computer vision”) AND TS = (“post-traumatic stress disorder” OR PTSD)). The timeframe for the data ranged from the earliest publication in this field (1999) to 31 December 2023, with an article, review article, proceeding paper, or early-access document. Duplicate and irrelevant articles (including those beyond the specified timeframe and non-English publications) were excluded. Ultimately, 431 relevant original articles were retained and exported as plain text files in the “Full Record and Cited References” format. This dataset can be used for bibliometric analysis and literature measurement purposes.

### 2.2. Data Analysis Methods

For our study, we selected three key tools: (i) VOSviewer (Version 1.6.19) (accessible at https://www.vosviewer.com/) serves as a graphical user interface software designed for co-citation and co-author analysis ([14]), accessed on 12 March 2024. (ii) CiteSpace (Version 6.3.R1 64-bit Basic) (accessible at https://citespace.podia.com/) is a Java-based tool that provides cluster and timeline views ([19]), accessed on 12 March 2024. (iii) Bibliometrix R-package (Version 4.1) (available at https://www.bibliometrix.org) is an open-source software developed in R, giving all the instruments to pursue a complete bibliometric analysis ([3]), following the Science Mapping Workflow, accessed on 15 March 2024.

Based on the bibliometric analysis, we hope to elucidate the development trajectory, current research status, publication information, and interrelationships in the field of AI in PTSD over the past 24 years, as well as to predict future trends. First, we extracted information from the WoSCC database and imported it into the Bibliometrix software for an overall analysis. This includes examining the past research development path and the number of articles published annually and their citations over the period from 1999 to 2023, as well as analyzing these trends. Second, we utilized software such as VOSviewer and Citespace to analyze co-citation networks of authors, institutions, and countries, aiding in establishing the current research status in this field. We further delved into keyword co-occurrence clustering, thematic evolution, and burst terms to identify research hotspots and future trends in the role of AI in PTSD. Detailed software usage and settings can be referenced in the framework (Figure 3).

## 3. Results

### 3.1. Overall Trend of Publications and Citations Analysis

As shown in Figure 4, the number of published articles on AI in PTSD demonstrates an overall upward trend. Based on the number of papers and their growth rate, the research period can be divided into three stages: the embryonic stage (1999–2006), the steady growth stage (2007–2014), and the rapid growth stage (2015–2023).

(i) Embryonic stage: During this period, there were only a few studies in this field in the WoSCC database, with the annual growth in articles not exceeding two. The average citation count in 2000 reached its highest point, indicating the initial attention given to this field.

(ii) Steady growth stage: In the first five years of this stage, the number of papers grew slowly. However, after 2012, the growth rate of publications increased, signaling broader attention to this field. This surge could be attributed to significant breakthroughs in AI driven by deep learning (DL) algorithms promoting machine learning (ML) development.

(iii) Rapid growth stage: From 2017 onwards, the number of papers increased substantially each year, peaking at 74 in 2022. This surge might be related to the COVID-19 pandemic starting in 2020 ([87]), where social isolation and the shift of medical resources brought practical attention to remote digital healthcare. However, the citation rate has decreased year by year.

### 3.2. Author Collaborative Network Analysis

Figure 5 is divided into two interconnected parts. The left network, based on the Price formula, identifies the core authors contributing to the field of AI’s role in PTSD and their collaboration patterns. The right side shows the authors’ production over time, highlighting the top ten authors by publication volume and their research duration in this field. The authors’ collaboration network comprises 296 nodes and 690 links. The size of each node reflects the author’s level of contribution. Node colors range from purple to red, indicating the research timeline from past to present, with line colors corresponding accordingly. The thickness of the lines denotes the strength of collaboration between authors, with thicker lines indicating stronger connections.

Overall, not all authors on the right side appear with labeled posts in the left network, suggesting that some authors, despite publishing many articles, need to improve their quality. Locally, the left network is divided into several smaller collaborative networks with strong internal cooperation but weak external connections. This suggests that authors in the field of AI in PTSD should broaden their collaboration to foster collective progress. Additionally, there is no dominant node in the network, with the largest component (1 CC) being 29 (9%), indicating the absence of a clear leader in the field. However, a few notable scholars include Kerry J. Ressler (Emory University, Atlanta), Jennifer S. Stevens (Emory University School of Medicine, Atlanta), and Katharina Schultebraucks (NYU Grossman School of Medicine). Although they have been active in the field for five years or less, they have achieved significant accomplishments. Since 2018, the number of contributing authors in this field has increased significantly, likely due to advancements in technology and growing market demand.

### 3.3. Contribution of Institutions and Countries Analysis

Figure 6 illustrates the global publication output and collaboration among countries. The legend indicates that the varying colors on the map, ranging from dark blue to light purple, represent different quantities of publications, with darker colors indicating higher numbers of publications. The United States ranks first with a publication volume seven times that of the second place, Canada (1282:181), followed by China (122). Additionally, the lines on the map represent the collaboration network between countries; thicker and darker lines indicate stronger collaboration relationships. The collaboration network is calculated using the formula Bcoll = A′ × A function, where nodes represent authors’ countries and links represent co-authors’ countries, showing collaboration frequency. The highest collaboration frequency is 16, found between the United States and China, followed by a frequency of 10 between the United States and Canada, Germany, and Israel. The United States also collaborates frequently with the United Kingdom, the Netherlands, and Australia, each with a frequency of 9.

This demonstrates that the United States plays a leading role in research on AI in PTSD. Canada, China, Australia, New Zealand, and several Western European countries (including the Netherlands, Germany, the United Kingdom, and France) have also formed several significant nodes. Notably, Israel, ranked thirteenth in terms of the number of published articles (6 papers), has an Average Article Citation (AC) ranking of 20 (tied for seventh with the United Kingdom), surpassing China, which is ranked third in terms of the number of publications (26 papers, AC = 13.9). This indicates that the quality of Israel’s articles is very high, possibly related to the country’s ongoing social environment of war and conflict, which will be discussed in detail in the discussion section. Meanwhile, the quality of academic articles from China needs improvement.

Based on Figure 7, we can identify the top eight institutions in terms of the number of publications in the field of AI’s role in PTSD from 1999 to 2023, as well as the relationships between these institutions, the top ten authors, and countries by publication volume. The institutions are as follows:

Harvard University (USA, 111 papers, research years: 2011–2023).

Veterans Health Administration (USA, 86 papers, research years: 1999–2023).

US Department of Veterans Affairs (USA, 77 papers, research years: 1999–2023).

Western University (Canada, 60 papers, research years: 2005–2023).

University of California System (USA, 51 papers, research years: 2003–2023).

Harvard Medical School (USA, 51 papers, research years: 2003–2023).

University of Minnesota Twin Cities (USA, 42 papers, research years: 2011–2023).

University of Minnesota System (USA, 35 papers, research years: 2006–2023).

Overall, collaboration is extensive, with almost all institutions having connections with multiple countries and individual authors collaborating with multiple institutions. Additionally, among the top 25 institutions, only Western University (ranked 4th, Canada), Tel Aviv University (ranked 20th, Israel), and the University of Sydney (ranked 22nd, Australia) are not based in the USA. This further confirms the observation that the USA places significant emphasis on research in this field and has substantial research outputs. Nonetheless, the academic influence of Canada and Australia is notable, with AC values of 44.0 and 30.3, respectively, surpassing that of the USA (28.2), which has the highest publication volume.

### 3.4. Publications Analysis

We applied Bradford’s Law using Bibliometrix to analyze the productivity of various journals. By using the principles of Bradford’s Law, Bibliometrix helps researchers determine the most active and influential journals in their dataset, providing valuable insights into the distribution of articles across journals in a particular research domain. This analysis led us to identify core journals in the field of AI’s role in PTSD, as shown on the right side of Figure 8. Among the 257 journals included in this study, 21 were considered core journals. The top five journals are the *Journal of Affective Disorders* (ISSN: 0165-0327), the *European Journal of Psychotraumatology* (ISSN: 2000-8198), *Frontiers in Psychiatry* (ISSN: 1664-0640), the *Journal of Psychiatric Research* (ISSN: 0022-3956), and *BMC Psychiatry* (eISSN: 1471-244X). It is evident that these journals are all related to psychiatry. And the left side of Figure 9 displays the sources’ production over time, highlighting the publication volume of the top seven journals in the field over the years. Literature growth rates for all journals are significantly higher after 2020.

Figure 9 displays the co-citation source network in the field of AI’s role in PTSD, as well as the impact data of the source on the right side. “Co-citation” refers to instances where the same or multiple publications are simultaneously cited by other documents. Co-citation analysis involves systematically studying these co-citation relationships to reveal key themes, research directions, academic influence, and other important information in a research field. The number of co-citations a journal receives affects the size of its node. A link represents an undirected connection or relation between two nodes, and thicker lines denote stronger co-citation relationships. Additionally, based on the similarity and co-occurrence relationships between nodes, three clusters were formed. Nodes within each cluster have high relevance, while nodes between different clusters are relatively independent. These clusters are represented by pink, blue, and cyan, indicating the three different journal groupings. The core representative journals for each cluster are the *American Journal of Psychiatry* (ISSN: 1535-7228), *Biological Psychiatry* (ISSN: 0006-3223), *NeuroImage: Clinical* (ISSN: 2213-1582), and *Archives of General Psychiatry* (ISSN: 0003-990X). These clusters correspond to traditional behavioral psychiatry, biomedical research, and data collection, respectively, representing the three different developmental directions of AI’s role in PTSD. The complex collaboration network demonstrates that the role of AI in PTSD is an interdisciplinary and collaborative field.

Based on the information on the right side of the figure, we can gain further insights into the scientific impact of the journals. The bottom right corner of the figure details the specific calculation methods used. The H-index, proposed by physicist Jorge E. Hirsch in 2005, is an indicator used to assess the academic output and impact of scientists, scholars, or research institutions, evaluating both the quantity and quality of academic achievements produced by research journals. The m-index aims to mitigate the influence of the duration of a researcher’s career on the H-index; a higher m-index typically indicates that the journal consistently produces highly impactful research within the field over time. The g-index considers the contribution of highly cited articles to the overall impact, aiming to provide a more comprehensive measure of a journal’s scientific influence. A larger g-index suggests that the journal not only has a significant number of highly cited papers but also includes some exceptionally highly cited papers. Journals with high values are highlighted with a pink background, including *Human Brain Mapping* (ISSN: 1065-9471), *Frontiers in Psychiatry* (ISSN: 1664-0640), *NeuroImage: Clinical* (ISSN: 2213-1582), *Journal of Affective Disorders* (ISSN: 0165-0327), *Biological Psychiatry* (ISSN: 0006-3223), and *Psychological Medicine* (ISSN: 0033-2917). Among these six journals, three are related to biological brain sciences, aligning with current trends in psychiatric research. This development is moving toward biomedical research, following the biopsychosocial model toward multidisciplinary comprehensive exploration.

### 3.5. Hotspots and Research Trends Analysis

#### 3.5.1. Co-Occurrence and Clustered Keyword Analysis

The keywords in our research paper aim to quickly and intuitively reflect the main content of the paper, thereby revealing the research hotspots and correlations of AI’s role in PTSD. We generated a keyword co-occurrence network diagram to explore the frequency and relationships between keywords. The left side of Figure 10 contains numerous nodes and connecting lines, where the size of the nodes represents the frequency of keyword occurrences, and the thickness of the lines indicates the strength of the association between two words. The colors show that the keywords are divided into three clusters, with a high correlation within each cluster.

The first cluster is red-purple, representing biomedical-related keywords. It includes terms related to brain structures, such as “functional connectivity,” “amygdala,” and “medial prefrontal cortex,” as well as disease states like “responses” and “emotion.” There are also technologies used for detection, such as “fMRI,” which play roles in prevention, diagnosis, neural mechanism analysis, and treatment effect evaluation. The second cluster, in cyan, represents keywords related to symptoms and causes, such as “comorbidity” and “veterans”. The third cluster, in blue, represents keywords related to machine applications. Keywords like “depression” and “anxiety” suggest that current AI tools are more often used in general psychological disorders. This is due to the high prevalence of these disorders and the variety of standardized assessment tools available, such as the Beck Depression Inventory (BDI) and Beck Anxiety Inventory (BAI), providing abundant data for AI training and analysis. Enhancing standardization in the diverse pathological conditions of PTSD is an area that needs more attention.

On the right side of Figure 10 is a timeline and frequency chart of the themes represented by the keywords. “Chronic pain” and “personality” have appeared in the trends of the past three years. Chronic pain often coexists with PTSD and shares some neurobiological mechanisms, such as chronic stress response, inflammatory response, and neurotransmitter changes. AI can conduct comprehensive analysis by processing and integrating data from various sources (such as electronic medical records, imaging data, and genetic data) ([105]), focusing on the comorbidity mechanisms and interactions between these two conditions. This is also part of personalized treatment. The keywords “veteran” and “childhood sexual abuse” have persisted for over 15 years, indicating that childhood sexual abuse is a significant trigger for PTSD. Individuals who experience childhood sexual abuse have a higher incidence of PTSD in adulthood, with severe and widespread consequences affecting the victims’ psychological, emotional, and social functioning.

Using CiteSpace’s Log-Likelihood Ratio (LLR) clustering algorithm, a cluster analysis of keywords in the field of AI role in PTSD was conducted. Cluster analysis can provide insights into the classification structure and research focus of a particular research topic based on citation relationships. The modularity value was Q = 0.7907, and the weighted mean silhouette value was S = 0.9142, indicating a clear division of co-citations and high homogeneity within clusters. Clusters are numbered based on size, with the largest cluster designated as Cluster #0.

According to Figure 11, “#0 Functional Connectivity” is the largest cluster, with internal nodes related to the biopsychosocial model of psychiatry (e.g., “amygdala”, “prefrontal cortex”) and methods of AI applied to psychiatry (e.g., “deep learning”). The second-largest cluster is “#1 risk factors,” associated with the causes of PTSD, including multiple keywords related to epidemiology. “#2 vietnam veterans” represents a large group of individuals with PTSD, reflecting social contexts and leading research from the United States, forming the third-largest cluster. Clusters #3, #7, #8, #10, and #11 are related to AI technologies, featuring various keywords about causes, symptom descriptions, algorithm logic, and coping strategies. Clusters #4-6 pertain to the pathological research of PTSD.

#### 3.5.2. Thematic Evolution Analysis

Thematic evolution analysis is a flexible qualitative method that explores the evolution patterns of themes and identifies persistent themes, aiding in understanding the interrelationships of phenomena. As shown in Figure 12, we divided the research on AI’s role in PTSD into three distinct sub-periods based on the number and growth rate of published papers in the WoSCC database. Each sub-period is represented by a separate chart: (a) 1999–2006, (b) 2007–2014, and (c) 2015–2023.

In the chart, each circle represents a theme, with the most frequently used keywords displayed above it. The size of the circle corresponds to the frequency of the keyword’s usage. Circles are distributed across different quadrants of the chart, divided based on the themes’ centrality and density, indicating the research status of different themes. Centrality measures the extent of external connectivity of a specific theme with other clusters, while density measures the degree of association between keywords, assessing the cohesion within the cluster.

The first quadrant (upper right) contains motor themes, which are widely recognized and highly important. The second quadrant (upper left) includes niche themes, which are specialized but have not been extensively researched. The third quadrant (lower left) features emerging or declining themes, a polarized area where themes are either newly developing or in decline. Themes that develop relatively slowly are categorized here, indicating a need for further in-depth research. Finally, the fourth quadrant (lower right) contains basic themes, which are characterized by high centrality and low density, suggesting that the connections between related keywords require further investigation.

As shown in Figure 12, the first sub-period (2006–2012) represents the preliminary exploration stage. During this period, research primarily focused on exploring the potential applications of AI technology in PTSD, including the potential of ML and Artificial Neural Networks (ANNs) in the diagnosis of mental health. The themes were specialized but not deeply investigated, indicating initial interest from researchers in this field. AI-related keywords such as “positron emissiontomography” appear in the upper right corner, representing early AI applications in brain imaging, indicating that AI’s role in image analysis was widely recognized and important during this period. This recognition is related to the continuous maturation of imaging technologies such as fMRI and PET.

In the second sub-period (2013–2018), representing the stage of expanded application, research themes became more diverse and were characterized by initial validation. Keywords like “neural networks” and “prefrontal cortex” in the upper right corner indicate widespread recognition of AI in auxiliary analysis related to recent rapid advancements in neuroscience, particularly in molecular genetics, neurobiology, and immunology. These advancements suggest new research trends. Keywords like “visual stimuli” and “antipsychotic” appear in the upper left corner, suggesting that methods for visually recreating traumatic events were explored, advancing research on using VR, XR, and other technologies to simulate virtual environments. The keyword “algorithm” in the lower left corner shows that many algorithms associated with PTSD, DL, and CNN were introduced for more accurate diagnosis and classification of PTSD. However, these technologies have not yet become widely applied or fully mature. During this phase, AI technology was not only used to develop wearable devices or mobile applications for real-time monitoring and intervention for patients but also to create personalized treatment plans based on patients’ historical data and real-time monitoring.

In the third sub-period (2019–2023), representing the stage of mature application, key themes became concentrated. Keywords like “post-traumatic stress disorder” and “meta-analysis” in the upper right corner became central themes, indicating that AI’s role in PTSD was widely recognized and researched, entering the stage of systematic reviews and analysis with a substantial accumulation of data and research results. For instance, research on brain function and structure was conducted through the meta-analysis project ’Enhancing Neuroimaging Genetics through Meta-analysis’ (ENIGMA). The keyword “data-base” on the left side indicates that data analysis became an important emerging research direction, although there remains significant room for improvement in underlying algorithm research. Keywords like “depression” and “symptoms” in the lower right corner are related to the heterogeneous nature of PTSD symptoms, emphasizing the importance of recognizing possible comorbid symptoms. During this phase, advancements in AI technology enabled the integration of multimodal data, including text, images, audio, and physiological data, to enhance the accuracy of diagnosis and treatment.

In summary, the figure illustrates the continuous advancement of AI technology in PTSD research, evolving from basic research to practical applications and now to integrated applications and ethical discussions. The development of AI algorithms and neuroimaging techniques has enabled a deeper exploration of brain biomedical science. Future research trends are likely to focus more on the integration of multimodal data, the optimization of personalized treatment plans, and the comprehensive analysis of social impacts ([83]).

#### 3.5.3. Hotspot and Frontier Analysis

Burst terms are important indicators for predicting sudden shifts in research directions or emerging trends in a specific field. Citation burst detection reveals the dynamic evolution of publications within a given time frame. By analyzing the duration, intensity, and burstiness of these terms, scholars can gain insights into the nature of these phenomena within the field. A sudden increase in citations of specific keywords or articles can indicate new discoveries and directions.

This section explores the citation burst detection of keywords related to AI’s role in PTSD research. Figure 13 lists the top 20 most cited keywords, providing insight into the recurring and influential themes in this field. “Prefrontal cortex” is the keyword with the longest burst duration, indicating it has been a research hotspot for the longest time. The keyword with the highest burst intensity is “neural network” (9.79), followed closely by “combat veteran” (8.33), showing a significant increase in citations during their highlighted periods, which occurred in 2015 and 2002, respectively. This reflects a trend from research focused on external environmental factors affecting specific populations to studies of internal neurobiological processes.

The keyword with the longest citation burst is “cerebral blood flow,” which lasted 13 years starting in 2004. This interest is related to theories proposed about the relationship between biogenetics and specific mental disorders, as well as the popularization of new technologies like functional magnetic resonance imaging (fMRI) and Positron Emission Tomography (PET). In recent years, burst keywords have increasingly diversified into more specific research areas. The most recent burst keyword is “Alzheimer’s disease”, which is linked to the increased incidence of Alzheimer’s due to long-term stress and trauma ([18]), as well as the aging population.

## 4. Discussion

### 4.1. Summary of Research Findings

The literature on AI’s role in PTSD has experienced significant growth since 2015, driven by advancements in AI technology, increased demand for telemedicine due to the COVID-19 pandemic, global economic downturns, and the frequency of recent wars. The author’s collaboration network indicates that cooperation is primarily concentrated within small groups, suggesting that the scope of collaboration in this field needs further expansion. The United States is the central country in this domain, forming extensive collaboration networks with scholars and institutions in countries such as China, Canada, and Australia. Although Israel has a relatively small number of publications, the quality of its papers is high, as reflected in a higher average citation count.

Research on AI in PTSD is inherently interdisciplinary, requiring close collaboration among experts from various fields. Computer scientists and AI specialists focus on algorithm development and optimization, while psychologists and psychiatrists provide clinical insights and behavioral analysis. Medical researchers concentrate on the exploration of biomarkers and clinical applications, public health experts assess the societal impact of large-scale applications, and ethicists and sociologists ensure fairness and ethical compliance throughout the research process. Data scientists and statisticians handle complex data and validate the effectiveness of AI models. Multidisciplinary collaboration not only drives technological innovation but also ensures the practical applicability of research outcomes in clinical and societal contexts.

Research hotspots are centered on the application of AI in PTSD, particularly in areas such as prediction and prevention, diagnosis and classification, treatment and prognosis, patient self-management, personalized interventions, and drug development. The integration of AI with neuroimaging, deep learning algorithms, and biological psychiatry models constitutes key themes in this field.

### 4.2. Analysis of Key Findings

#### 4.2.1. The U.S.’s Leading Position and Innovation-Driven Model in AI for PTSD Research

The United States boasts world-class universities and research institutions, as discussed in Section 3.3, with examples provided. It is also home to leading technology companies such as Verily (Alphabet’s life sciences division), IBM (IBM Watson Health), and Google (Google Health). These institutions are global leaders in fields like AI, computer science, medicine, and psychology. The U.S. government places high importance on research in AI and PTSD, driving rapid advancements through policy support and funding. For instance, the Defense Advanced Research Projects Agency (DARPA) has backed neuroscience research leveraging AI technologies through initiatives like the “Next Generation Nonsurgical Neurotechnology (N3)” and the “Brain-Computer Interface (BCI)”. These projects aim to develop non-invasive techniques to improve treatment and rehabilitation for PTSD patients. Additionally, the passage of the OPEN Government Data Act has made vast amounts of healthcare data available for researchers, while the National Institutes of Health (NIH) supports cutting-edge neuroscience research through the BRAIN Initiative, fostering the use of AI to enhance diagnostic tools and personalized treatments. Adequate research funding and advanced infrastructure position the U.S. at the forefront of AI and PTSD research.

The U.S. also has access to extensive datasets and faces a broad spectrum of healthcare needs. Due to its involvement in wars such as those in Vietnam and Afghanistan, the Department of Veterans Affairs (VA) and the Department of Defense (DoD) have accumulated substantial data on the mental health of military personnel and veterans, including a wealth of PTSD-related data ([89]). Additionally, the U.S. has conducted large-scale mental health surveys, such as the National Comorbidity Survey (NCS) and the National Epidemiologic Survey on Alcohol and Related Conditions (NESARC) ([44]), which provide valuable information on PTSD prevalence, associated factors, and comorbidity with other mental health issues. For example, the lifetime prevalence of PTSD among U.S. adults is approximately 7% ([81]).

The U.S. follows a multi-level integrated innovation-driven model that combines government policy support, data-driven initiatives, collaboration between tech companies and academic institutions, interdisciplinary cooperation, continuous technological innovation, and the translation of research into practical applications. This comprehensive approach offers a valuable model for other countries to learn from.

#### 4.2.2. The Reasons Behind Israel’s High-Quality Advances in AI for PTSD Research

Israel’s complex military background, marked by long-standing security challenges, has resulted in a higher prevalence of PTSD among military personnel and veterans. This specific context drives the military’s commitment to providing substantial funding and resources, supported strategically by the government, to produce high-quality research aimed at addressing these real-world issues. Notably, there is a focus on leveraging AI technologies for early diagnosis, treatment optimization, and prevention of PTSD.

Israel is home to several of the world’s leading research institutions, such as Tel Aviv University (ranked 20th in this field), Hebrew University, and the Weizmann Institute of Science. These institutions have a strong research foundation in artificial intelligence and mental health. Despite its relatively small academic scale, Israel is actively involved in international collaborations, partnering with top scholars and institutions in the U.S., Europe, and beyond. Additionally, many Israeli tech companies are heavily invested in applying AI to healthcare research, including PTSD diagnosis and treatment, and they work closely with academia to enhance the quality of their research.

#### 4.2.3. The Importance of Neuroimaging

Neuroimaging employs various imaging techniques to observe and measure the brain’s structure, function, blood flow, and metabolic activity. It is crucial for understanding normal brain function, mental disorders, brain injuries, and neurodegenerative diseases ([75]; [98]). With the aid of AI technology, neuroimaging can swiftly identify abnormalities, detect characteristic changes in the brains of PTSD patients, predict treatment outcomes, and deepen the understanding of pathological mechanisms.

In the neural circuitry model of PTSD, the hypothalamic–pituitary–adrenal (HPA) axis plays a crucial role in stress response and cortisol regulation ([41]). The activation of the amygdala, prefrontal cortex, and hippocampus stimulates the HPA axis ([88]; [8]), leading to the secretion of adrenaline and cortisol. However, high concentrations of cortisol can persist in humans for several days to months. Prolonged elevated cortisol levels can cause cell death in the hippocampus, resulting in hippocampal shrinkage ([46]). This explains why PTSD patients have an impaired ability to terminate stress responses, difficulties in storing new positive memories, and challenges in dissociating past trauma from the present reality. Additionally, there is reduced ([75]; [98]) to less engagement in deep cognitive processing and more direct responses from the amygdala ([49]; [26]). The amygdala becomes hyperactive in response to traumatic stimuli, directly perceiving emotions and triggering responses, which may cause patients to exhibit more defensive, passive, or aggressive behavior. Different types of trauma are associated with specific amygdala subregions and their volumes. Meanwhile, the prefrontal cortex, which is essential for emotional regulation, shows lower resting cerebral blood flow in PTSD patients compared to healthy individuals ([16]) Studies also indicate that enhanced immune status, elevated pro-inflammatory cytokines, and increased arachidonic acid metabolites of COX-2 contribute to the neurobehavioral deterioration observed in PTSD ([31]). These subtle differences, ranging from the molecular to the systemic level, also help explain why the prevalence of PTSD is higher in women compared to men ([6]).

Neuroimaging tools include MRI, fMRI, structural MRI, diffusion MRI, PET, EEG ([47]), CT, and NIRS. Among these, fMRI is commonly used to observe the morphological and functional changes in the brains of PTSD patients, predict treatment responses, and aid in formulating personalized treatment strategies.

However, the clinical application of neuroimaging faces challenges, such as high technical demands, complex sequences, and issues with accuracy and reproducibility ([51]). Most studies involve small sample sizes (usually fewer than 300 participants), and the complexity of comorbidity issues leads to inconsistent data and varied results. Additionally, most studies rely on standardized spatial analysis, which may overlook crucial individual differences in brain function and structure.

In the future, the development of neuroimaging will focus on individualized precision imaging and AI-based big data analysis. By using personalized approaches to describe differences in brain function and structure and exploring their impact on behavior, researchers can better understand the unique needs of PTSD patients and develop corresponding treatment plans. More precise analysis with AI technology is expected to improve diagnostic accuracy and advance the field of precision medicine.

### 4.3. Advantages and Challenges of AI in PTSD

#### 4.3.1. The Practical Advantages of AI in PTSD Applications

The practical significance of AI technology in the diagnosis and treatment of PTSD is highly notable. Firstly, AI enables early diagnosis of PTSD through efficient and accurate screening tools, which is particularly beneficial for high-risk populations such as veterans and disaster survivors, allowing them to receive timely interventions. Secondly, AI leverages patient-specific data to assist clinicians in developing personalized treatment plans and optimizing therapeutic outcomes. Additionally, AI tools can provide auxiliary diagnostic support in resource-limited settings, reducing healthcare costs and alleviating the shortage of mental health professionals ([52]). Ultimately, these applications contribute to mitigating the socio-economic burden of PTSD and advancing global mental health management and policymaking.

AI has demonstrated significant efficacy in clinical trials, especially in PTSD diagnosis. For instance, Liu et al. employed SVM with resting-state functional magnetic resonance imaging (rs-fMRI) data to differentiate PTSD patients from healthy controls with an accuracy of 92.5%, sensitivity of 90%, and specificity of 95%, highlighting AI’s potential for individual-level PTSD identification ([55]). Similarly, Zhang et al. integrated rs-fMRI and structural magnetic resonance imaging (sMRI) data using a multi-kernel SVM approach, achieving a classification accuracy of 89.19% between PTSD patients and healthy controls, further validating AI’s value in enhancing diagnostic performance ([107]). Gong et al. demonstrated AI’s role in detecting structural brain abnormalities in PTSD by comparing gray and white matter volumes, achieving an accuracy of 91% in distinguishing PTSD patients from healthy controls ([104]).

In the assessment and monitoring of PTSD treatment, AI algorithms offer innovative digital therapeutic approaches that can significantly alleviate or even eliminate patient symptoms with minimal side effects. For instance, Freespira, a digital therapeutic developed by Freespira, helps patients by regulating breathing patterns and has received FDA approval ([103]). Another example is NightWare, which helps PTSD patients achieve more restful sleep by monitoring and interrupting traumatic nightmares. It has completed randomized controlled trials that validate the effectiveness of its therapy ([63]). The development and application of these digital tools provide new treatment options for PTSD patients, particularly for those who have limited access to traditional clinical services.

These cases underscore the immense potential of AI technology in improving the accuracy, sensitivity, and specificity of PTSD diagnosis, offering new directions for future clinical diagnostics and treatments. These research findings not only reinforce the utility of AI in advancing early screening efficiency but also pave the way for its broader application in PTSD management.

#### 4.3.2. Data Collection Barriers and Bias

In the field of AI in PTSD, the integration of AI algorithms with big data plays a crucial role. A notable example of this is the collaboration between Epic Systems, the world’s largest electronic health record (EHR) system, and OpenAI, a leader in generative AI technology. This partnership explores how to integrate generative AI into EHR systems to enhance automation and data analysis in clinical documentation ([27]). For instance, GPT models can be employed to generate clinical notes, automatically populate medical records, or assist physicians in summarizing patients’ medical histories ([84]).

However, the collection of medical data faces significant challenges and barriers. Concerns about data privacy and security lead many patients and healthcare institutions to be highly cautious about sharing or storing data on third-party platforms. Legal regulations over the years, such as the European Union’s General Data Protection Regulation (GDPR) in 2015, have established complex access controls to protect data. These regulations necessitate that patients may need to individually consent to data-sharing requests from different healthcare providers, and institutions must obtain explicit patient consent before collecting and using their data, adding operational complexity and increasing management costs ([54]).

Data collection, storage, and management require specialized information technology infrastructure and professionals, which poses difficulties, especially in remote and technologically underdeveloped regions where inadequate hardware and software facilities hinder data collection ([15]). These challenges are further exacerbated by the emergence of “data silos”, where data are often stored in separate, often competing, hospitals, clinics, and laboratories without standardized technical protocols or interoperability agreements, making data sharing and integration difficult ([93]).

To address this, there should be a push toward the standardization of medical data formats and protocols, along with the development of incentives and mechanisms for data sharing among healthcare institutions. Technologies such as federated learning and differential privacy could be introduced to ensure data privacy while enabling effective data utilization ([78]; [65]). Additionally, optimizing relevant legal frameworks to streamline consent management processes and investing in more infrastructure to enhance the accessibility of AI-based digital healthcare is essential.

Furthermore, biases in the data used for training, the objectives of model optimization, and the subjective judgments of developers during algorithm design can lead to systematic biases in the outcomes or decisions ([9]). This may result in certain groups receiving unfair treatment when accessing PTSD-related medical resources. If the public becomes aware of this, it could reduce societal acceptance of the AI industry as a whole, thereby impacting clinical feasibility and potentially leading to legal disputes. This underscores the need to further enhance the fairness and transparency of AI systems.

#### 4.3.3. Socio-Economic Impact of AI in PTSD Diagnosis and Treatment

The widespread adoption of AI technology in the clinical application of PTSD holds significant socio-economic value but is influenced by factors such as technological maturity, the effectiveness of early medical practice, and, most critically, ethical and social acceptance. While AI-assisted medical scenarios, such as triage, pre-diagnosis, and medical record generation, have demonstrated the potential to enhance patient experiences, reduce the risk of misdiagnosis, and improve healthcare quality ([1]), skepticism persists among some patients due to misconceptions about the ethics and reliability of AI. This highlights the necessity of increasing public education and awareness through visual aids, lectures, and other formats to break down cognitive barriers and foster trust. Additionally, user-friendly and accessible platforms can enhance patient acceptance and satisfaction, addressing resistance to technology.

Beyond social acceptance, the integration of AI technology in PTSD treatment has broader implications for reducing the socio-economic burden of the disorder. PTSD often leads to decreased productivity and increased long-term medical expenses, placing a significant strain on families and society. By improving diagnostic accuracy and treatment outcomes, AI can reduce healthcare costs, facilitate early recovery, and help patients reintegrate into society and the workforce ([79]). This not only underscores AI’s vast potential in addressing the global mental health crisis but also paves the way for its incorporation into clinical guidelines and the advancement of technology-driven healthcare policies, ensuring a more sustainable and effective allocation of medical resources.

### 4.4. Limitations of the Research

This study only accessed the WoSCC database, and despite identifying 984 overlapping articles on AI’s role in PTSD across WoSCC, PubMed, and Scopus, some articles may still have been missed. Additionally, due to selection bias in the literature review, we made judgments on the type and language of the literature, only considering studies published in English, which may have led to the omission of relevant research published in other languages. In terms of analysis methods, we only used CiteSpace, VOSviewer, and Bibliometrix for analysis and did not conduct more in-depth analyses such as meta-analysis. Therefore, we performed content analysis in Section 4.1 and Section 4.2 to mitigate the inherent limitations of bibliometric conclusions and to clarify the relationships and causes within the data. Other limitations in research on AI’s role in PTSD include sample representativeness and insufficient consideration of follow-up evaluations. The remote nature of AI interventions can make it difficult to monitor and ensure patient adherence ([95]), which may affect the validity of clinical studies.

### 4.5. Future Research Directions

#### 4.5.1. Transparent and Fair Algorithm Design

In the future, the development of transparent and fair algorithms will require a comprehensive approach, considering multiple factors. First, algorithmic fairness refers to the principle that ML algorithms should make decisions without bias or discrimination against any group or individual (e.g., based on gender or race), thus avoiding unfair predictive outcomes ([23]; [35]; [21]). To achieve this, it will be necessary to clarify and standardize the definitions and metrics of algorithmic fairness, establishing appropriate evaluation criteria and benchmarks. For example, common fairness criteria in binary classification tasks—such as demographic parity, predictive equality, and equal opportunity—should be effectively integrated into the design and implementation of algorithms.

Additionally, the future will see the development and use of more advanced bias mitigation techniques, including preprocessing steps (e.g., importance weighting and resampling), in-processing steps (e.g., incorporating non-discrimination constraints into models), and post-processing steps (e.g., adjusting outputs to meet group fairness metrics) ([56]). However, the issue of dataset shift could lead to inconsistent algorithm performance across different subgroups, exacerbating healthcare disparities. Understanding these health disparities and inequalities is crucial for assessing the fairness of AI models across diverse subgroups ([73]).

Furthermore, improving algorithmic interpretability will be essential in medical decision-making and model auditing, enabling a better understanding of the sources of unfairness and the detection of dataset shifts. For future developments, federated learning can be adopted as a distributed learning paradigm ([20]), allowing the use of local data and computational resources to collaboratively train a global model without sharing sensitive information. Additionally, employing fair representation learning can help reduce the unfair impact of algorithms on specific groups ([20]). Enhancing model interpretability, researching ways to make decision-making processes more transparent, and providing clear explanations to healthcare professionals and patients will be vital for building trust in AI systems.

Importantly, increasing the diversity of datasets to ensure that algorithms perform well across different groups (e.g., race, gender, age) is a key measure. Moreover, establishing regulatory frameworks and ensuring compliance is crucial. For instance, following guidelines set by the U.S. Food and Drug Administration (FDA) can help ensure the fairness and transparency of AI software as a medical device (AI-SaMD) ([22]).

Another critical aspect of researching algorithmic fairness is fostering interdisciplinary collaboration in the future. Experts in data statistics, computer science, and policy-making must work together to advance the research and practice of algorithmic fairness. Continuous monitoring and evaluation are essential to ensure that algorithms perform fairly in real-world scenarios, allowing for feedback-based adjustments and optimizations. Through improvements in technology, policy, and organizational structures, the fairness and transparency of AI in PTSD can gradually be enhanced, better serving the needs of society and users.

#### 4.5.2. System Compatibility with Multimodal Data Integration

In the field of PTSD, AI-powered multimodal data fusion systems demonstrate immense potential, and their current limitations point to several future research directions ([97]). Multimodal machine learning, which integrates various data types (such as text, images, audio, and video) ([91]), achieves cross-perception fusion, thereby endowing machines with enhanced comprehensive perception and understanding capabilities. Currently, different modalities are somewhat isolated; however, developing a systematic multimodal framework could significantly improve accuracy. Feature extraction is a core aspect of multimodal data processing. By utilizing autoencoder networks for preprocessing and feature extraction of multimodal data, these features can be fused into a primary shared representation, providing more comprehensive information and improving model performance. This approach not only enhances data expressiveness but also enables the model to better understand and address complex real-world problems. Dynamic Neural Fields (DNFs) also present a promising option, where methods integrating prior knowledge adjust weights to prioritize more effective modalities, further boosting overall accuracy.

In the future development of multimodal disease detection, end-to-end cross-modal machine learning pipelines will play a crucial role. These pipelines can standardize the representation of different modalities and achieve more precise diagnostic and treatment plans through knowledge-guided data fusion.

Areas such as virtual reality and behavioral interventions, emotion recognition and empathetic AI, and multimodal emotion analysis and affective computing also hold significant potential for advancement. These research directions are expected to significantly enhance the diagnosis and treatment of PTSD, providing patients with more accurate and personalized support.

#### 4.5.3. Deep Integration with Traditional Medical Practices

Future research should focus on how to better integrate AI into existing hybrid systems to enhance overall treatment outcomes. There is a strong need for increased interdisciplinary collaboration, particularly between neuroscience and AI, to address similar neurodegenerative diseases such as Alzheimer’s disease or other prevalent and unresolved complex conditions like midlife depression and childhood autism ([101]). Specifically, attention should be given to how AI can be deeply integrated into traditional PTSD treatment systems. This integration could lead to a collaborative healthcare ecosystem where AI complements the diagnostic work of human doctors by providing support in areas such as auxiliary diagnosis, real-time monitoring, and telemedicine ([45]; [76]) while also reducing ethical controversies. It is crucial to establish clear regulations and legal frameworks to ensure stricter oversight of data privacy and ethics, thereby promoting the standardized development of AI in the medical field, including PTSD.

## 5. Conclusions

Based on bibliometric analyses of publications from 1999 to 2023, the integration of AI into PTSD clinical practice offers significant promise for enhancing diagnostic accuracy, personalizing treatment and enabling real-time patient monitoring. Current authors’ collaborations are often limited in scope, but future research could benefit from broader partnerships. The United States leads in this field with its multi-tiered, innovation-driven model, which could serve as a valuable reference for other countries.

Advanced AI algorithms, such as ML, DL, and multimodal data fusion systems, have the potential to address many challenges across the PTSD care continuum and facilitate more in-depth research into this heterogeneous disorder. Implementation methods include mobile application software, wearable devices, various virtual reality technologies, and even brain–computer interface technologies. Moreover, in remote areas with scarce medical resources, AI can effectively mitigate the consequences of a lack of specialized personnel and facilities.

However, to fully realize AI’s potential, several key challenges must be addressed. These include data privacy concerns, conflicts between extensive data collection and analysis, algorithmic biases, and the need for robust IT infrastructure, particularly in resource-limited settings. Ethical implications of AI in healthcare also need careful consideration, with the establishment of transparent frameworks and regulations to ensure responsible use. Successful integration of AI into PTSD treatment requires interdisciplinary collaboration, especially between AI experts and neuroscientists, and the development of standardized protocols for seamless integration into existing healthcare systems. By tackling these challenges and fostering interdisciplinary cooperation, AI has the potential to significantly improve treatment outcomes for PTSD patients and pave the way for its application in other complex mental health conditions.

## Figures and Tables

**Figure 1 behavsci-15-00027-f001:**
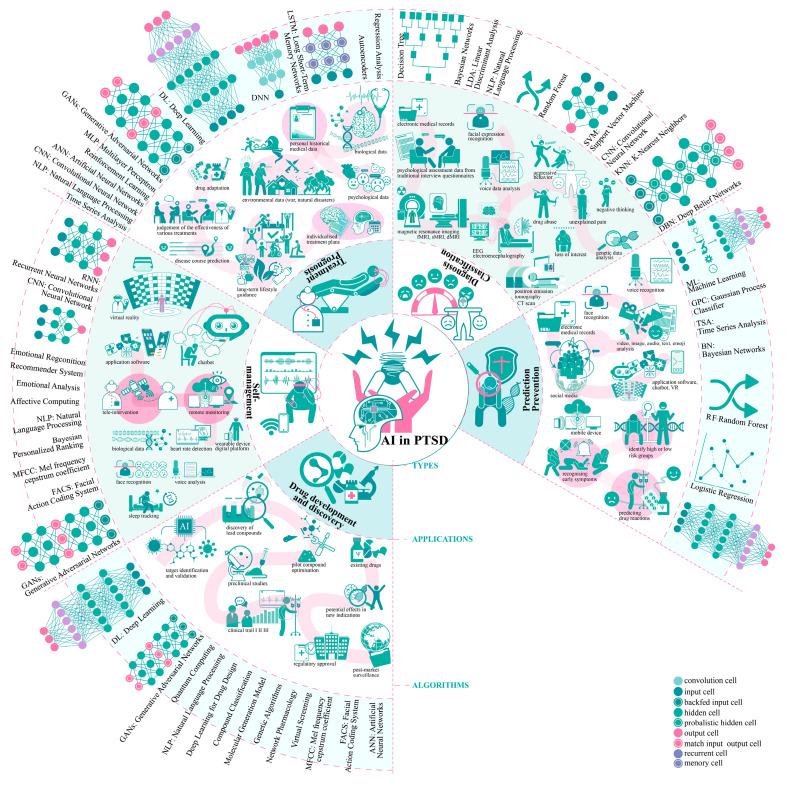
The types and applications of AI in PTSD visualization.

**Figure 2 behavsci-15-00027-f002:**
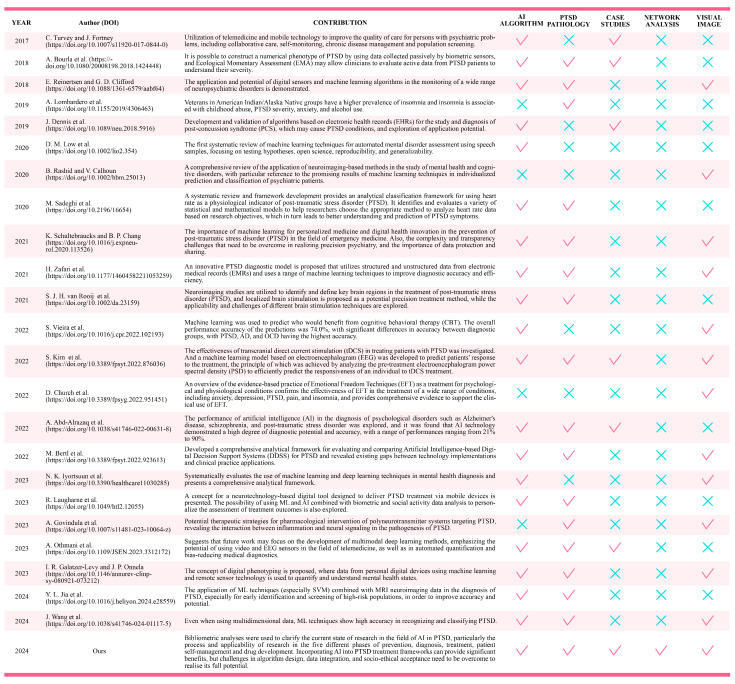
Summary of related works versus our survey ([95]; [11]; [76]; [57]; [27]; [59]; [75]; [80]; [83]; [105]; [98]; [100]; [46]; [24]; [1]; [9]; [39]; [51]; [31]; [69]; [29]; [40]; [102]).

**Figure 3 behavsci-15-00027-f003:**
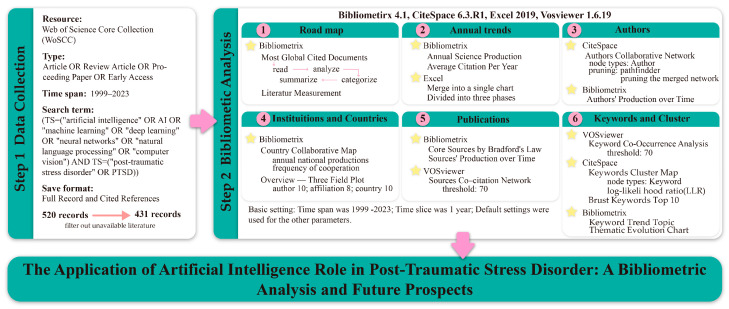
Bibliometrix analysis framework.

**Figure 4 behavsci-15-00027-f004:**
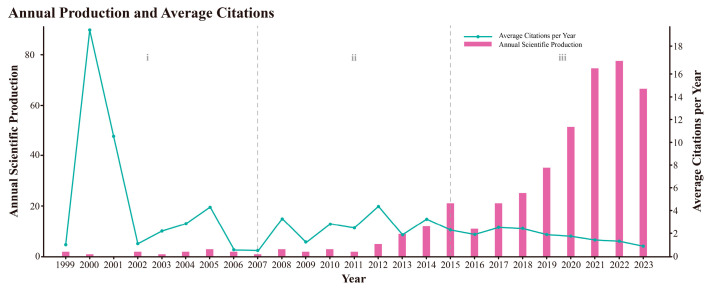
The overall trend of publications and average citations. The annual trend of publications and average citations per article on research on the application of AI in the diagnosis and treatment of PTSD indexed in the Web of Science Core Collection (WoSCC) database from 1999 to 2023 using Bibliometrix.

**Figure 5 behavsci-15-00027-f005:**
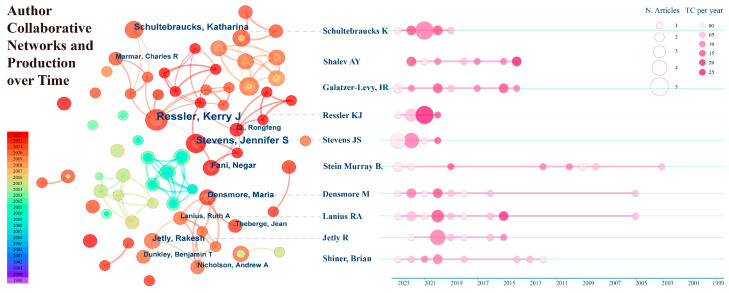
The overall trend of publications and average citations. The annual trend of publications and average citations per article on research on the application of AI in the diagnosis and treatment of PTSD indexed in the Web of Science Core Collection (WoSCC) database from 1999 to 2023 using Bibliometrix.

**Figure 6 behavsci-15-00027-f006:**
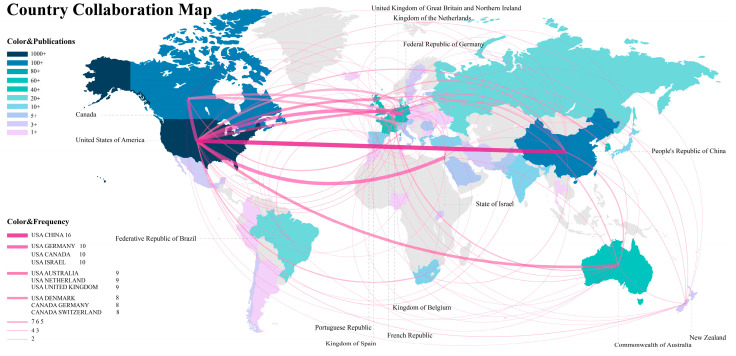
Country collaboration map, with a publication threshold of 1 and a collaboration threshold of 2. This figure was created using Bibliometrix.

**Figure 7 behavsci-15-00027-f007:**
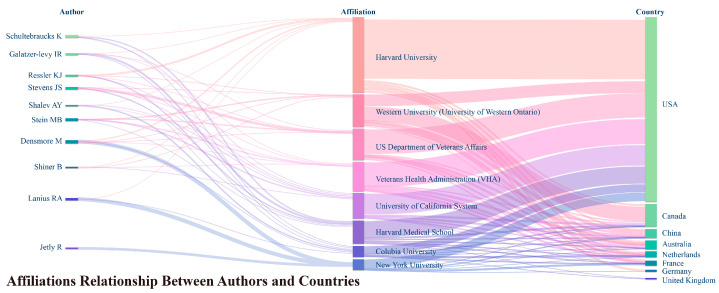
Affiliation relationship between authors and countries. Overview tree-field plot of affiliations (items 8), authors (items 10), and countries (items 10) using Bibliometrix.

**Figure 8 behavsci-15-00027-f008:**
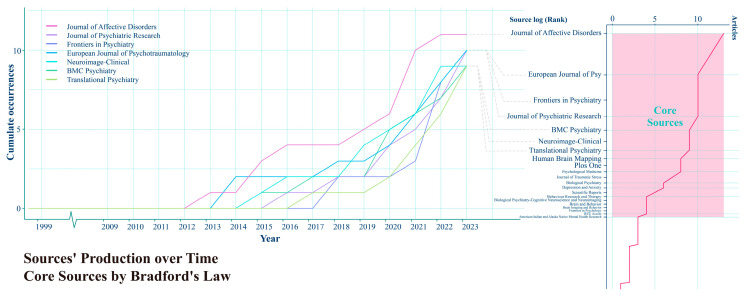
Journal productivity and core journal distribution analysis. This figure was created using Bibliometrix.

**Figure 9 behavsci-15-00027-f009:**
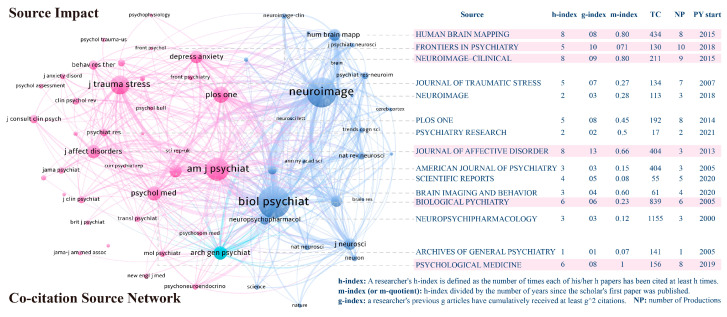
Publication co-citation network with impact information. This visualization is based on the “co-citation” file type, with the unit set to “cited sources”; among the 5708 data points, we set a minimum citation threshold of 70. A total of 62 journals exceeded this threshold, resulting in 62 nodes. This analysis was conducted using Vosviewer. The sources’ impact information analysis was performed using Bibliometrix.

**Figure 10 behavsci-15-00027-f010:**
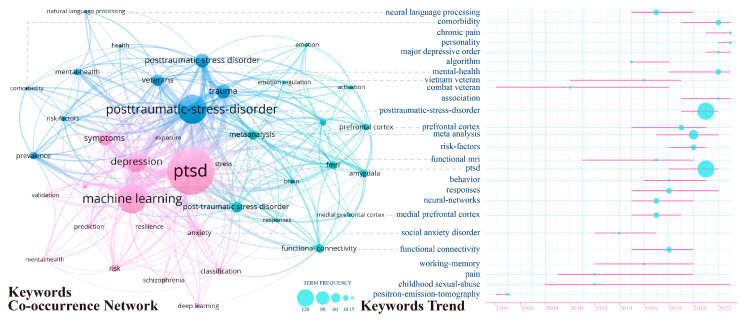
Keyword co-occurrence network and trend. Select the “Co-occurrence” file type and set the unit to “All Keywords”. Out of 2230 keywords, we set the minimum occurrence threshold to 15, resulting in 39 keywords meeting this criterion using VOSviewer. The keyword trend timeline was created using Bibliometrix.

**Figure 11 behavsci-15-00027-f011:**
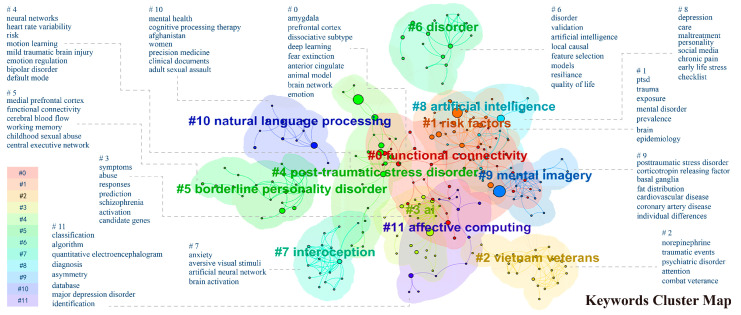
Keyword cluster map. The keyword clustering analysis considered all keywords, using the g-index (k = 10) as the selection criterion, and resulted in 11 clusters based on data from 431 documents. These clusters form the basic components of Figure 10, with each cluster having its own color and label. This figure was created using CiteSpace.

**Figure 12 behavsci-15-00027-f012:**
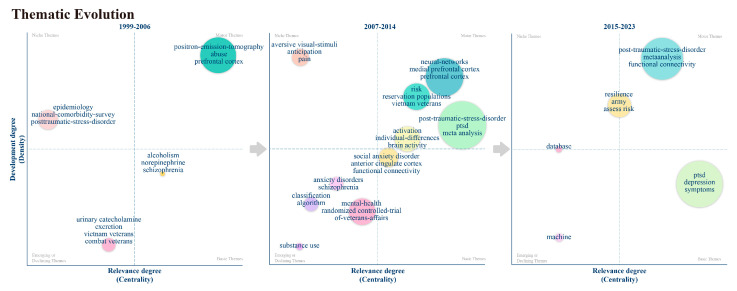
Thematic evolution map. This map illustrates the different states of research themes in the field of AI’s role in PTSD across three distinct periods: (i) 1999–2006, (ii) 2007–2014, and (iii) 2015–2023. This figure was created using Bibliometrix.

**Figure 13 behavsci-15-00027-f013:**
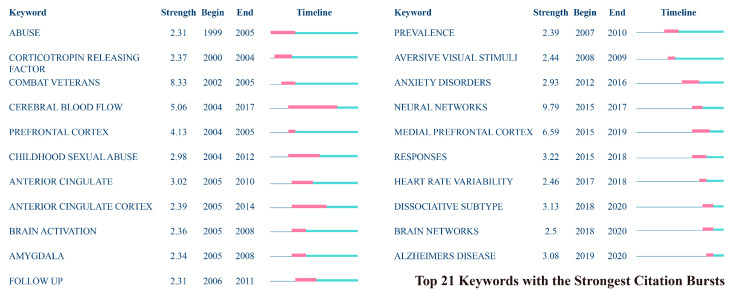
The top twenty-one keywords with the strongest citation bursts using CiteSpace.

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
