# Peer review of "Current Status and Future Directions of Artificial Intelligence in Post-Traumatic Stress Disorder: A Literature Measurement Analysis"

_behavsci, 2024, doi:10.3390/bs15010027_

Round 1
Reviewer 1 Report
Comments and Suggestions for Authors
Dear Authors!
Your study, entitled: Current Status and Future Directions of Artificial Intelligence in Post-Traumatic Stress Disorder: A Literature Measurement Analysis, addresses an important, relevant, current scientific question. The study's genre is review, which precisely meets the purpose of the article. "This study aims to explore the current state of research and the applicability of artificial intelligence (AI) at various stages of post-traumatic stress disorder (PTSD)"
The introduction of the study is appropriate, accurately introducing the mental illness PTSD and its causes in modern people (subsection 1.1.2). The study clearly outlines how artificial intelligence technology has emerged in relation to mental illnesses; It precisely details the areas: "early detection, and predictive models, AI provides individuals with more accurate evaluation tools and the ability to identify early signs of mental health issues. Additionally, AI can tailor personalized treatment plans based on individual characteristics and offer immediate support through virtual therapists and chatbots, overcoming temporal and geographical barriers. AI’s continuous monitoring capabilities also enable real-time tracking of patients".
Then it presents, analyzes and visualizes all of this in a list format (subsection 1.1.5, Figure 1).
The methodology chosen for the review is appropriate, with the help of bibliometric analysis, the authors of the study can get answers to their research questions. The methodology of bibliometric analysis is also presented precisely and professionally in subsection 1.2.2.
The precise description of the research goal is also omitted: "This paper aims to systematically explore the current research status and development trends of AI in the field of PTSD through bibliometric analysis. It analyzes the collaboration networks of authors, institutions and countries, identifies key contributors and
their relationships, and provides valuable insights for future research, as well as data support for research management, funding decisions, and resource allocation".
I do not consider the description and visualization of the structure of the study (Figure 3) to be necessary or important, since the text follows the structure of scientific writing and study writing exactly anyway.
Anyway, the structural structure of the paper is logical, proportional, and complies with the rules of scientific writing.
The data sources and data used during the research are also described in detail in subsection 2.1.
The presentation of the results is of a good standard, it precisely follows the research questions: Overall Trend of Publications and Citations Analysis, Author Collaborative Network Analysis, Contribution of Institutions and Countries Analysis Publications Analysis, Hotspots and Research Trends Analysis. The results are also visualized (Figure 5-12) and in relation to the 3 sub-periods of the longitudinal study (2006-12, 2013-18, 2019-23). ​​The division of the sub-periods is logical, the results are supported by arguments and visualized (Figure 13.).
During the discussion, the authors accurately analyze the results obtained.
The presentation of the limitations of the study and the necessary directions for further research is good. The emphasis on ethical issues is particularly important: " it is crucial to prioritize the development of standardized protocols for AI implementation, foster interdisciplinary collaboration—especially between AI and neuroscience—and address public concerns about AI's role in healthcare to enhance its acceptance and effectiveness. "
The list of cited literature is adequate.
Reviewer
Author Response
Thank you very much for your thorough and thoughtful review of our manuscript entitled "Current Status and Future Directions of Artificial Intelligence in Post-Traumatic Stress Disorder: A Literature Measurement Analysis." We greatly appreciate the time and effort you dedicated to reading our study and providing such detailed and constructive feedback.
We are pleased to hear that you found our study relevant, clear, and well-structured. Your comments highlight several aspects of our work that we aim to improve further.
Please find the detailed responses below and the corresponding revisions have been highlighted in red color in the re-submitted files.
Once again, thank you for your time and thoughtful feedback. We believe that the revisions we have made in response to your comments significantly improve the manuscript. We hope that the revised version meets your expectations, and we look forward to your continued guidance as we move forward with the publication process.

Reviewer 2 Report
Comments and Suggestions for Authors
The article aims to explore the current state of research on the use of AI in the management of post-traumatic stress disorder, from prevention to treatment development and patient self-management.
- The article addresses a highly relevant topic, considering the prevalence of PTSD in diverse contexts.
- The methodology is solid. A bibliometric analysis with advanced tools is used, which ensures a robust quantitative approach to the analysis of literature. The identification of trends based on keywords and collaborative networks adds value.
- Particularly interesting is the visualization of the information. Without a doubt, the graphs and diagrams offer clarity in the evolution of topics, collaborative networks and trends, useful for both researchers and policy makers.
- It highlights current limitations such as biases in algorithms, the need for larger samples and the importance of addressing ethical concerns
- the depth in clinical analyses could be improved. Although specific applications of AI in diagnosis and treatment are mentioned, including more clinical details or case studies demonstrating the tangible impact of these technologies would enrich the study.
All in all, the article is a valuable contribution to the field, offering a broad overview and clear visualizations on the use of AI in PTSD. The work could benefit from a more balanced approach combining bibliometric analysis with practical cases.
Author Response
Thank you very much for taking the time to review this manuscript. Please find the detailed responses below and the corresponding revisions/corrections highlighted/in track changes in the re-submitted fileThank you for your thorough and thoughtful review of our manuscript entitled "Current Status and Future Directions of Artificial Intelligence in Post-Traumatic Stress Disorder: A Literature Measurement Analysis." We appreciate the time and effort you dedicated to evaluating our work. Your feedback provides valuable insights that have helped us improve the clarity and depth of our study.
We are pleased that you found the topic of AI in PTSD management highly relevant and that you acknowledged the solid methodology and the effectiveness of the visualizations we presented.
Once again, we are grateful for your valuable feedback, which has helped us improve the manuscript and ensure that it provides a well-rounded and comprehensive view of AI’s role in PTSD research and treatment. We hope that the revisions meet your expectations and look forward to your further comments.
